# Determination of Thrombogenicity Levels of Various Antiphospholipid Antibodies by a Modified Thrombin Generation Assay in Patients with Suspected Antiphospholipid Syndrome

**DOI:** 10.3390/ijms23168973

**Published:** 2022-08-11

**Authors:** Pavla Bradáčová, Luděk Slavík, Adéla Skoumalová, Jana Úlehlová, Eva Kriegová, Gayane Manukyan, David Friedecký, Barbora Piskláková, Jana Ullrychová, Jana Procházková, Antonín Hluší

**Affiliations:** 1Masaryk Hospital Usti nad Labem, Department Clinical Hematology, 40113 Usti nad Labem, Czech Republic; 2Department of Hemato-Oncology, Faculty of Medicine and Dentistry, University Hospital Olomouc, Palacky University Olomouc, 77900 Olomouc, Czech Republic; 3Department of Internal Medicine III-Nephrology, Rheumatology and Endocrinology, Faculty of Medicine and Dentistry, University Hospital Olomouc, Palacky University Olomouc, 77900 Olomouc, Czech Republic; 4Department of Immunology, Faculty of Medicine and Dentistry, University Hospital Olomouc, Palacky University Olomouc, 77900 Olomouc, Czech Republic; 5Laboratory of Molecular and Cellular Immunology, Institute of Molecular Biology NAS RA, Yerevan 0014, Armenia; 6Laboratory for Inherited Metabolic Disorders, Faculty of Medicine and Dentistry, University Hospital Olomouc, Palacky University Olomouc, 77900 Olomouc, Czech Republic

**Keywords:** antiphospholipid syndrome, thrombosis, thrombogenicity, seronegative APS, anti-cardiolipin, anti-β2-glycoprotein-I, anti-phosphatidylserine/prothrombin, anti-annexin V, ELISA, chemiluminescence analysis, thrombin generation assay, lupus anticoagulants, FV Leiden heterozygous

## Abstract

Antiphospholipid syndrome (APS) is a hypercoagulable state accompanied by the presence of heterogeneous antiphospholipid antibodies (aPL), which nonspecifically affect hemostasis by the presence of lupus anticoagulans (LA), anticardiolipin antibodies (aCL), antibodies against β2-glycoprotein-I (anti-β2GPI), but also non-criteria antibodies such as antibodies against β2-glycoprotein-I domain I (anti-DI), anti-phosphatidylserine/prothrombin (anti-PS/PT), anti-annexin V, and many others. The main target of the antibodies is the activated protein C (APC) system, the elimination of which can manifest itself as a thrombotic complication. The aim of this study was to determine the thrombogenicity of antibodies using a modified protein C-activated thrombin generation assay (TGA) on a group of 175 samples suspected of APS. TGA was measured with/without APC and the ratio of both measurements was evaluated (as for APC resistance), where a cut-off was calculated ≤4.5 (90th percentile) using 21 patients with heterozygous factor V Leiden mutation (FV Leiden heterozygous). Our study demonstrates the well-known fact that multiple positivity of different aPLs is a more severe risk for thrombosis than single positivity. Of the single antibody positivity, LA antibodies are the most serious (*p* value < 0.01), followed by aCL and their subgroup anti-DI (*p* value < 0.05). Non-criteria antibodies anti-annexin V and anti-PT/PS has a similar frequency occurrence of thrombogenicity as LA antibodies but without statistical significance or anti-β2GPI1 positivity. The modified TGA test can help us identify patients in all groups who are also at risk for recurrent thrombotic and pregnancy complications; thus, long-term prophylactic treatment is appropriate. For this reason, it is proving increasingly beneficial to include the determination antibodies in combination with modified TGA test.

## 1. Introduction

Antiphospholipid syndrome (APS), also referred to as Hughes syndrome, was first described in 1983 by Dr. Graham Hughes [1]. APS is an autoimmune disease accompanied by persistent antiphospholipid autoantibodies (aPL). The main bindings targets of aPL are phospholipid membranes of platelets and their subsequent activation, but also endothelia, monocytes, and neutrophils with a procoagulant effect [2,3]. All of these can subsequently lead to thrombosis [4].

The prevalence of aPL in the population is approximately 1–5%, but only a minority develop APS [5]. The clinical manifestations of APS are highly variable; venous thrombosis can manifest as phlebothrombosis or pulmonary embolism. Arterial thromboses often result in myocardial infarction or stroke. In pregnancy complications, APS is often the cause of pre-eclampsia, spontaneous abortion, premature birth, or fetal death.

The criteria for APS according to the Sydney classification are very well-defined; at least one clinical and at least one laboratory criterion must be met. The clinical criteria for APS include the presence of arterial or venous thrombosis and reproductive loss [6,7]. Laboratory criteria include positivity of at least one lupus anticoagulant (LA) antibody, anti-cardiolipin (aCL) IgG, IgM, anti-β2-glycoprotein-I (anti-β2GPI) IgG, IgM [8]. To meet the laboratory criterion, aPL must be repeatedly positive within a 12-week [9]. Single, double, or triple positivity is evaluated, as patients with triple positivity have the highest risk of thrombosis and recurrent miscarriages [10,11,12].

The different antibodies investigated in APS (LA, aCL, anti-β2GPI, anti-prothrombin (anti-PT), anti-phosphatidylserine/prothrombin (anti-PS/PT), and anti-phosphatidylethanolamine) can be evaluated for specificity of phospholipid surface inhibition. However, laboratory evaluation of antibody manifestations is difficult, and thus, most studies are unfortunately based on testing the clinical manifestation APS of a specific antibody. The combination of LA + anti-β2GPI + anti-PS/PT provides the best diagnostic accuracy for APS, according to Scascia [13].

From the point of view of the antibody isotype, the most important for the clinical manifestation of APS are LA with anticoagulant activity, in contrast to the isolated aCL and anti-β2GPI in the IgG and IgM classes. The importance of the IgA class is also rarely described.

Regarding the pathophysiology, a number of possible activation mechanisms of antiphospholipid antibodies against cellular targets (monocytes, platelets, endothelial cells) have been described; however, activated protein C (APC) appear to be the main target in the humoral part. Protein C (PC) is activated by thrombin bound to thrombomodulin and serves as a key regulator of thrombosis by inactivating FVa and FVIIa and activating thrombin receptor on the surface of platelet, which leads to the inhibition of apoptosis. Modulation of APC increased resistance to APC and increased thrombin generation. The presence of in vitro APC resistance is associated with the presence of LA and a clinical history of thrombosis.

The pathogenic mechanisms by which aPL induce disease expression in APS are manifold and the corollary of this is the exceptional variability in the clinical manifestations observed in these patients. Unfortunately, the molecular mechanisms that lead to aPL development and aPL-induced disease expression remain incompletely understood. Methods that can be used to detect these changes are molecular-based, flow-cytometric, immunological, and functional. However, they do not fully reflect the in-vivo action of antibodies on target molecules and need to be complemented by clinical outcomes [14].

The aim of our study was to evaluate the hypercoagulability state in APS after a short discontinuation of anticoagulant with both thrombin generation (TGA) profiles and activated protein C resistance (APCR) in different types of APS. In contrast to basic coagulation assays such as prothrombin time and activated partial thromboplastin time (aPTT), which measure thrombin generation only during the initiation phase of coagulation, TGA measures thrombin generation during the entire clotting process and thus provides a broader picture of the overall coagulation potential. This property of TGA analysis can be used to detect various causes of hypercoagulable states; when Douxfils et al. [15] used a modification of TGT with the addition of APC to measure the effect of hormonal contraceptives, it was also possible to use this modified test to determine the influence of phospholipid antibodies to APC.

### 1.1. Antiphospholipid Antibodies

There are a variety of antiphospholipid antibodies that interact with the negatively charged phospholipid surfaces of many cells and tissues by various mechanisms. These aPLs include the APS criteria antibodies lupus anticoagulans, anti-cardiolipin, anti-β2-glycoprotein-I, and the APS non-criteria antibodies anti-β2-glycoprotein-I domain I, anti-phosphatidylserine/prothrombin, anti-annexin V, and others.

#### 1.1.1. Seronegative APS

In practice, we not infrequently find patients with clinical manifestations of APS but who are repeatedly negative for all criteria antiphospholipid antibodies. These are the so-called seronegative APS (SN-APS) patients [16,17,18,19]. Some SN-APS patients show repeated positivity of non-criteria antibodies such as anti-DI, anti-PS/PT IgG, IgM, anti-annexin V IgG, IgM, and others [20,21,22].

#### 1.1.2. APS Criteria Antibodies

Lupus anticoagulants are a heterogeneous class of immunoglobulins that specifically target epitopes of the negatively charged cell membrane phospholipid-binding protein, prothrombin, and beta2-glycoprotein I (β2GPI), which prolong phospholipid-dependent coagulation tests in vitro when competition with coagulation factors for phospholipid binding occurs [23]. LA positivity, compared to aCL and anti-β2GPI, as well as other non-criteria antibodies, is a much higher risk factor for thromboembolism, cerebral ischemia, and recurrent reproductive loss [24].

Anti-cardiolipin antibodies include a group of antibodies directed against the cardiolipin portion of the veneral disease research laboratory (VDRL) antigen, which are antibodies that react with the phospholipids of the prothrombin activator complex and antibodies capable of reacting with cardiolipin in the solid phase [25]. Also, aCL IgG is much more strongly associated than aCL IgM with cerebral thromboses and myocardial infarctions. Detection of aCL can be performed by chemiluminescence assay (CLIA) and enzyme-linked immunosorbent assay (ELISA). The established cut-off (99th percentile) for ELISA for aCL positivity is >40 GPL/MPL [8]. For CLIA, the manufacturer’s recommended cut-off for aCL positivity is >20 CU (99th percentile) [26].

Anti-β2-glykoprotein-I is an anionic glycoprotein with five phospholipid-binding domains. Four domains have regular, conserved sequences, but the fifth domain is aberrant. This domain contains a six-residue insertion, a C-terminal extension of 19 residues, and an additional disulfide bond that includes a C-terminal cysteine. These additional amino acids in the V domain are responsible for the unique property of this CCP domain because they form a large positively charged patch that determines the affinity for anionic phospholipids [27]. The anti-β2GPI IgG, IgM antibody plays a major role in the pathogenesis of APS. Its presence is strongly associated with thromboembolic complications [28].

The β2-glycoprotein-I molecule is composed of five homologous domains and occurs in two conformations, either in a closed circular form or in an open form. In the circular form, interaction with anti-β2GPI occurs mainly between domains 1 and 5, whereas in the open form, an epitope is exposed at domain 1, to which anti-β2GPI binds. Detection of anti-β2GPI IgG, IgM is performed by ELISA according to the international recommendations of the Society of Thrombosis and Haemostasis Scientific and Standardization Committee ISTH SSC. The established cut-off (99th percentile) for ELISA for anti-β2GPI positivity is >40 GPL, or [8]. The anti-β2GPI IgG assay has higher specificity for APS than aCL IgG, but lower sensitivity for APS than aCL IgG assay [22]. However, anti-β2GPI results do not always correlate significantly with clinical manifestations of APS, which may be due to the lack of standardization of the ELISA method [29,30,31,32,33]. The modern method of anti-β2GPI detection is CLIA, the cut-off for positivity is >20 CU (99th percentile) [34].

#### 1.1.3. APS Non-Criteria Antibodies

The presence of APS anti-β2-glykoprotein-I domain I antibodies, compared to other aPLs, correlates much more strongly with the incidence of thrombosis and reproductive loss [35]. The presence of anti-DI together with LA is significantly associated in patients with APS and venous thrombosis [22]. Anti-DI positivity was significantly associated with triple positivity. In contrast, anti-DI negativity was significant in patients with isolated presence of other criteria aPL. The manufacturer’s recommended cut-off for anti-DI positivity in CLIA is >20 CU (99th percentile) [26,35].

Anti-phosphatidylserine/prothrombine complex IgG antibody may be a very useful predictive factor for thrombosis in patients with SLE [36]. Anti-PT is also capable of binding to the PS/PT complex. The positivity of anti-PS/PT IgG, IgM is highly significantly associated with arterial and venous thrombosis and pregnancy problems when LA positivity is present [37,38,39], and the sensitivity and specificity for APS is higher than that for aCL positivity [40]. Liu et al. using ELISA (cut-off > 30 [41]) detected anti-PS/PT IgG, IgM in 72% of APS samples, 36% of SN-APS, and 0% of healthy donors. The detection of anti-PS/PT was higher in both APS and SN-APS groups than the detection of aCL IgG, IgM and anti-β2GPI IgG, IgM. They further found that specifically anti-PS/PT IgG was the best predictor for deep vein thrombosis, OR = 9.2 [26,35].

Annexins belong to the group of Ca^2+^-dependent phospholipid-binding proteins. Annexin V is a major component of trophoblast and vascular endothelium. Annexin V binds phospholipids with anticoagulant activity, serving as a so-called protective shield. If annexin V interacts with antibodies, this shield can be disrupted, causing thrombosis and reproductive loss [42]. However, the correlation of anti-annexin V with pregnancy complications is not entirely significant and further studies are needed [43].

## 2. Results

The TGA assay was performed in two separate reactions with/without APC by modifying the putative antiphospholipid antibody target, which is activated protein C. To evaluate the effect of antibodies, the ratio of both determinations was used, which will allow to evaluate the effect of antibodies on APC without separate effects of the basic coagulation state in individual patients. Although all TGA parameters were measured, the effect of antibodies on the TGA-modified assay was evident especially in lag time and the total amount of thrombin generated (AUC). For each group of patients (175), healthy volunteers (47) and patients with FV Leiden heterozygous (21), the ratios between the two measurements were calculated for the parameters tLag, tPeak, tmPeak, and AUC: R = TGA − APC/TGA + APC.

Patients (175) were divided according to the positivity of each aPL into several groups: single positivity (108), double positivity (27), and triple positivity (4). To assess the severity of the risk of individual antiphospholipid antibodies, we divided patients with single positivity into other subgroups according to aPL positivity: single positivity LA (61), single positivity aCL (37), and single positivity anti-β2GPI (10). SN-APS patients (36) were divided into two subgroups: anti-PS/PT (9) and anti-annexin V (27). There were 70 females and 69 males in the APS patient cohort (139). In the SN-APS patient cohort (36) there were 14 women and 22 men. In the group of healthy volunteers (47) there were 34 women and 13 men. The individual data are shown in Table 1 and Table 2.

The parameters tLag R, tPeak R, and mPeak R do not show significant differences between the groups and are prolonged in all groups due to the presence of aPL; see Figure 1.

We have used the 21 patients with known hereditary thrombophilia with heterozygous mutation FV Leiden to determine cut-off for modified thrombin generation test, because there is no existence of any cut-off of evaluation of the risk of thrombogenicity. Significant differences between groups were found for the parameter AUC R = AUC − APC/AUC + APC. In patients with FV Leiden heterozygous, AUC R values ranged from 1.20–25.32. Based on these, a cut-off of ≤ 4.5 (90th percentile) was set for the AUC R parameter. The AUC R results are shown in Figure 2. Summary of the representation of patients with a cut-off ≤ 4.5 for AUC R in each group are shown in Table 3.

## 3. Discussion

The modified TGA method with the addition of APC was used to detect antiphospholipid antibodies, as in the work of Douxfils et al. [15], where it was used to determine the thrombogenicity of hormonal contraceptives. Our results show comparable interassay variability. On the basis of previously published results comparing the two available TGA methods (Wayne Chandler et al. [44]), our results can be advantageously adapted to a second routinely used testing platform. Expression of the results of the modified TGA method as a ratio R = TGA − APC/TGA + APC provides the possibility to evaluate the change in thrombin generation depending on the presence of APC and avoids the influence of interindividual variability caused by the presence of different types of antibodies. At the same time, by measuring the ratios, we avoid possible assay variability caused by the time factor of collection and other laboratory variables [44].

Since the modern adaptation of the thrombin generation test, interpreting the result of the cut-off determination for individual situations has been a fundamental question. Although a number of papers have been published on the use of TGA in clinical situations ranging from thrombogenicity to various conditions, through the effects of anticoagulant therapy to the risk of spontaneous bleeding, exact cut-off values for these conditions have not been established [45]. Some works have tried to do this, but the set values have not been widely used [46]. Determining the effect of antiphospholipid antibodies on TGA is a great challenge, as it is well-known that antibodies occupy phospholipid surfaces and thus prolong coagulation assays, but paradoxically create a thrombophilic state. One explanation may be the elimination of the protein C activation system, which also needs phospholipid surfaces for the activation step.

We determined four parameters: tLag R, tPeak R, mPeak R, AUC R. The parameters tLag R, tPeak R, and mPeak R do not show significant differences between the groups and are prolonged in all groups due to the presence of aPL. The parameter AUC R, cut-off ≤ 4.5 (90th percentile), was found to be the most conclusive for the comparison of thrombogenicity, showing significant differences between groups similar to the study by Billoir et al. [47].

The main objective of our study was to determine thrombogenicity antiphospholipid antibodies by the modified thrombin generation assay in group of 175 patients with clinical manifestation of APS with positivity of various antiphospholipid antibodies and 47 healthy volunteers, and then to compare the results with the thrombogenicity [48]. We used 21 patients with a well-known genetic thrombophilia with heterozygous factor V Leiden mutation to establish a cutoff value for the assessment of thrombogenic risk. A cut-off of ≤ 4.5 (90th percentile) was set for the AUC R parameter. Comparison of patients in our study according to risk of thrombosis; triple positivity (LA, aCL, anti-β2GPI), single positivity for each antibody (LA, aCL, anti-β2GPI, anti-annexin V, anti-PS/PT); and a lone group of patients with positivity anti-DI and simultaneity with the next one or more antibodies demonstrated thrombogenicity of individual antibodies to APC.

The higher measured frequency of occurrence thrombogenicity in the triple-positive APS group (cut-off was reached in 50.0% of patients) is consistent with the clinically increased incidence of thromboses in triple-positive patients when all three aPL (LA, aCL, and anti-β2GPI) are present simultaneously [10,12], but not statistically significant. The lupus anticoagulant antibody has a very strong association with thrombosis, having the most significant clinical impact in the development of thrombosis compared to all other aPLs [24]. This is confirmed by our study, as the group of patients with single LA positivity showed the second highest thrombogenicity (cut-off was reached in 41.0%) after the group of patients with triple positivity, even though patients with different LA positives from low to high positives (NR = 1.25–2.30) were included in the study. Patients with low positivity LA (NR = 1.2–1.5) numbered 55, patients with medium positivity LA (NR = 1.5–2.0) numbered 8, and patients with high positivity (NR > 2.00) numbered 2. Similar results were achieved by LiestØl et al., who showed that plasma samples from patients with positive LA showed resistance to the anticoagulant effect of APC [49].

In general, anti-β2GPIs are often cited as an important cause of APS, but direct clinical evidence is lacking. Jiang et al. [50] performed a systematic review of several medical databases and found only a very weak independent association of anti-β2GPI with thrombosis based on data from 4758 retrieved articles. This fact is confirmed by our findings, where a comparison of thrombogenicity in patients with aCL single positivity (cut-off was reached in 32.4%) and anti-β2GPI single positivity (cut-off was reached in 10.0%) shows that anti-β2GPI has a much lower thrombogenic potential than aCL antibody. The single anti-β2GPI frequency of occurrence thrombogenicity is the lowest of all the groups we studied, even lower than that of the group of healthy volunteers.

Anti-cardiolipin frequency of occurrence thrombogenicity is the fourth highest in our study. Based on a meta-analysis of 11 studies with a total of 2425 patients, Pastori et al. [51] showed a significant association between aCL positivity and coronary artery disease, especially in patients under 50 years of age. In patients with high aCL titers, the risk of a second myocardial infarction after the first myocardial infarction is at least twice as high. The APC-R value indirectly correlates with the aCL level.

Now we discuss the thrombogenicity of non-criterial antibodies (anti-annexin V, anti-PS/PT, anti-DI), which is often doubted. Anti-annexin V does not belong to the aPL criteria in APS according to Sydney recommendations [8], and therefore, even if patients meet the clinical manifestations of APS, they do not meet the laboratory criteria and are considered SN-APS. The group of patients with anti-annexin V positivity showed the third highest frequency of occurrence thrombogenicity (cut-off was reached in 40.7%). It is clear from clinical practice that the risk of thrombosis is increased in these patients, but there are still no official recommendations for prophylactic antithrombotic [52,53]. It would certainly be worth considering supplementing the Sydney laboratory criteria for APS with other aPLs, such as anti-annexin V. Also, for this reason, we consider it advisable to investigate TGA to assess the degree of thrombogenicity and the risk of thrombosis, at least in patients with positive findings of non-criteria antibodies.

In the group with positive anti-PS/PT 22.2% of patients reached a cut-off to determine frequency of occurrence thrombogenicity, which is twice as much as in the group of healthy volunteers however, statistically nonsignificant. This result may indicate a certain thrombogenic potential of anti-PS/PT. Zhu et al. [54] performed meta-analysis of 21 studies from a few scientific databases on occurrence of anti-PS/PT with a total of 1853 patients and found that anti-PS/PT alone occurs in up to 65% patients with clinical manifestation APS. Egri et al. [55] identified occurrence of anti-PS/PT in their study in 28% of patients.

Anti-DI positivity occurs predominantly with positivity of other aPLs, often in triple positive APS. Anti-DI alone is very rare. Patients in our study also have anti-DI positivity and one other aPL positive at the same time. The presence of anti-DI may increase the risk of thrombosis. Surprisingly, in our study, the frequency of occurrence thrombogenicity in anti-DI was only 16.7%, but it was statistically significant (*p* value < 0.05). Measurement of a larger number of patients will be needed to more accurately determine thrombogenic potential. In contrast, Slavik et al. [56] examined 74 patients with APS who had concurrent positivity in at least one aCL and anti-β2GPI class. They found anti-DI positivity in 21 samples, 57% of whom had clinical manifestations of APS. By performing anti-DI testing, they increased the predictive value for thrombosis from 25% to 68% in anti-DI positive patients. Similarly, Guo et al. [57] showed positivity anti-D1 was related to increased risks for thrombosis odds ratio 29.87 (95% confidence interval).

## 4. Methods and Materials

### 4.1. Study Design and Population

We detected 243 samples; of these, 175 were patients with clinical manifestations of APS according to Sydney criteria classification, that is, each patient had thrombosis or fetal loss and was positive for at least one aPL: LA, aCL, anti-β2GPI. In addition, 47 healthy volunteers as negative control group and 21 patients with positive activated protein C resistance in heterozygous carriers of the Leiden FV mutation (FV Leiden heterozygous) as positive control group were included in the retrospective study to determine the cut-off.

All patients (175) and healthy volunteers (47) were tested for criteria aPL: LA, aCL IgG, IgM, anti-β2GPI IgG, IgM, and anti-DI. Positivity of at least one aPL criteria was found in 139 patients. In 36 SN-APS patients with negative criteria aPL, the investigation was extended to determine other non-criteria aPL: anti-PS/PT IgG, IgM and anti-annexin V IgG, IgM. In the group of healthy volunteers (47), there was no positivity for either criterion aPL. Patients taking warfarin or heparin and patients with FV Leiden heterozygous were not included in the study. In patients taking direct oral anticoagulants (DOACs), drug elimination from plasma was performed with a DOAC stop tablet. Slavik et al. conducted a study to verify the effect of DOAC stop tablets and found that after elimination of the DOAC drug from the plasma sample, neither screening nor special coagulation tests were affected [58].

### 4.2. Sample Collection

Samples of non-clotting blood were collected into tubes containing the anticoagulant sodium citrate 3.2%. Double centrifugation at 2500× *g* was performed within 1 h after collection, immediately after which platelet-poor plasma (PPP) aliquots were frozen at −80 °C. Aliquots were thawed at 37 °C for 10 min before measurement.

### 4.3. Coagulation Tests for Lupus Anticoagulans

LA testing was performed based on the ISTH SSC international recommendation for the detection of Lupus anticoagulants [59]. Detection is based on the ability of antiphospholipid antibodies present in the patient’s plasma to prolong the coagulation time in a phospholipid-dependent assay. The following tests were performed on a Sysmex CS 5 100 coagulation analyzer (Sysmex, Kobe, Japan) using reagents: Diluted Russell Viper Venom Time (dRVVT) and aPTT (Siemens Healthineers, Erlangen, Germany). Measurements were performed according to the reagent kit manufacturer’s recommendation. For screening tests were used reagents with low concentration phospholipids and for confirmation reagents with high concentration phospholipids [60], in three basic steps: 1, screening; 2, mixing test; and 3, confirmation [61].

The LA results were interpreted as positive/negative according to the ISTH SSC recommendations based on the calculation of the normalized ratio (NR). The calculation of the NR is as follows: NR = R screening (time patient/time polled normal plasma)/R confirmation (time patient/time polled normal plasma). The cut-off value for NR is 1.2. Samples that exceeded the cut-off > 1.2 were determined to be LA-positive. [62]

### 4.4. Chemiluminiscence Immunoassay for aCL, Anti-β2GPI including Structural Variation Anti-DI

Examination of IgG, IgM and IgG, IgM and anti-DI (QUANTA Flash, Inova Werfen, Barcelona, Spain) was performed by quantitative chemiluminescence method on a Bioflash analyzer (Inova Werfen, Barcelona, Spain). Measurements were performed according to the reagent kit manufacturer’s recommendation. The cut-off recommended by the manufacturer is >20 U/mL [41,63,64].

### 4.5. Enzyme Linked Imunosorbent Assay for Anti-PS/PT and Anti-Annexin

The determination of anti-PS/PT IgG, IgM (QUANTA Lite, Inova Diagnostics Werfen, Barcelona, Spain) and anti-annexin V IgG, IgM (Orgentec Diagnostics, Mainz, Germany) was performed by ELISA using the Biosan instrument kit (Biosan, Riga, Latvia). Measurements were performed according to the reagent kit manufacturer’s recommendation. The cut-off given by the reagent manufacturer for the positivity of each aPL is as follows: anti-PS/PT IgG, IgM > 30 IU/mL, anti-annexin V IgG, IgM > 5 IU/mL [65,66,67,68].

### 4.6. Thrombin Generation Assay Modified by APC

Thrombogenicity was determined by the thrombin generation assay using the Technothrombin^®^ TGA assay and APC reagent (Technoclone, Vienna, Austria) on a Ceveron alpha (Technoclone, Vienna, Austria). The risk of bleeding or thrombosis can be assessed based on TGA testing. Based on an analysis of the literature, Tripodi et al. concluded that TGA is a promising laboratory tool for investigating bleeding coagulopathies and predicting the risk of recurrent venous thromboembolism [69,70]. Similarly, Binder et al. document in their review article most of the clinical applications of TGA for the prediction of thrombosis [71]. TGA measures the total hemostatic potential of plasma. In contrast to basic coagulation assays such as prothrombin time and aPTT, which measure thrombin generation only during the initiation phase of coagulation, TGA measures thrombin generation during the entire clotting process and thus provides a broader picture of the overall coagulation potential. The amount of thrombin is determined by the fluorogenic substrate that is cleaved by the generated thrombin. The rate of fluorescence is directly proportional to the amount of thrombin generated. The course of thrombin generation is shown in a curve in which the following four parameters are evaluated: tLag (min), time to the start of thrombin generation, reference range 2.3–5.1 min; tPeak (min), time from the start of thrombin generation to its maximum concentration, reference range 6.2–9.8 min.; mPeak (nM), maximum concentration of generated thrombin, reference range 21.7–214.8 nM; AUC (nM) = endogenous potential (ETP); total amount of generated thrombin, area under the curve, reference range 1.014–1.949 nM.

In our study, for each PPP sample, one TGA measurement was performed with reagent with added activated protein C (+APC) and a second measurement without added activated protein C (−APC). From both measurements, the TGA − APC/TGA + APC ratio was calculated for each parameter [47]. Thrombin generation in the sample was initiated with 7.16 pM of recombinant tissue factor (rTF) resuspended in 0.32 μM of phospholipid micelles (containing 2.56 μM of phosphatidylcholineand 0.64 μM of phosphatidylserine). For APC, activated protein C (16 nmol/L) was added.

### 4.7. Statistics

Statistical analysis was performed using the software GraphPad Prism 9.0 (GraphPad Software, San Diego, CA, USA). To normalize original (tLag R) data-log-transformation was applied. Consequently, Shapiro–Wilk test confirmed the gaussian distribution of all groups studied. Outliers were identified by Grubb’s test (alpha = 0.05). Data were visualized by scatter dot plots including median and quartiles. ANOVA with Dunnett’s multiple comparisons post-hoc test was applied to calculate statistical significance of differences between FV Leiden heterozygous and other groups. 

## 5. Conclusions

The modified assay allows for determining the sensitivity of TGA to one of the most potent inhibitory systems of coagulation, which is APC. The modified TGA assay with APC allows the ratio between measurements with and without APC to be evaluated, allowing the assay to be standardized. From statistical evaluation, the AUC R parameters and especially their ratio are most useful, but the take is influenced by the parameter tLag R. As there is no defined cut-off for the AUC R parameter, the determination of thrombogenicity was made against the group of FV Leiden heterozygous patients.

The following conclusions follow from the assessment of the individual groups. The criteria of antibodies LA, aCL, and anti-β2GPI are strongly associated with the clinical manifestations of APS. Other non-criteria antibodies, such as anti-phosphatidylserine/prothrombin and anti-annexin V, are also currently being studied and their role in the development of thrombotic complications in the patient [72]. However, the presence of these non-criteria aPL has not yet been shown in clinical research to significantly increase thrombogenicity and cause thrombosis.

Our study demonstrates the well-known fact that multiple positivity of different aPLs is a more severe risk for thrombosis than single positivity. Of the single antibody positivity, LA antibodies are the most serious (*p* value < 0.01). However, even anti-annexin V frequency of occurrence thrombogenicity has similar LA antibodies in patients, which contrasts with the low values achieved with anti-β2GPI1 positivity.

The modified TGT test can help us identify patients in all groups that are also at risk for recurrent thrombotic and pregnancy complications and for whom long-term prophylactic treatment is appropriate [22,28]. For this reason, it is proving increasingly beneficial to include the determination of antibodies in combination with modified TGA test [19]. The introduction of additional aPL as well as TGA measurements into laboratory practice could be a useful tool to refine and speed up the diagnosis of APS or the identification of patients at risk and the manifestation of thrombotic complications in APS. The subsequent initiation of antithrombotic therapy could lead to a reduction in the recurrence of thrombosis and pregnancy complications [73,74,75,76].

## Figures and Tables

**Figure 1 ijms-23-08973-f001:**
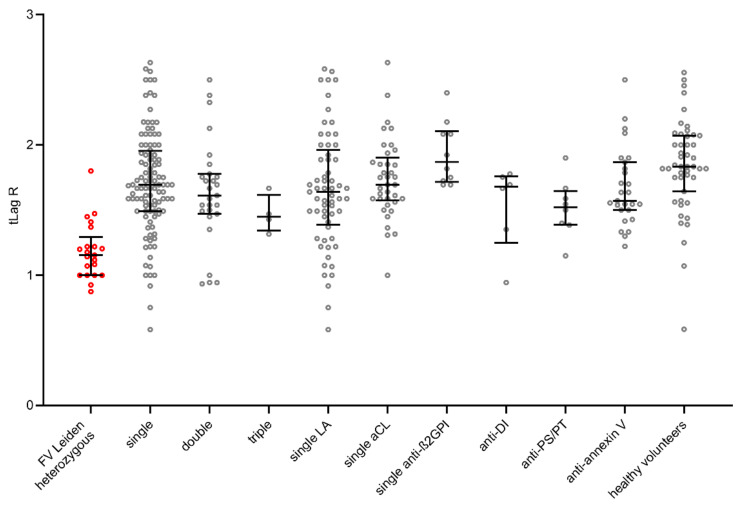
Visualization of TGA tLag R in each group by scatter dot plots including median and quartiles. TGA: thrombin generation assay; tLag R: lag phase 1/(tLag − APC/tLag + APC). The group of FV Leiden heterozygotes as a standard of thrombogenicity is shown by the red shaded points.

**Figure 2 ijms-23-08973-f002:**
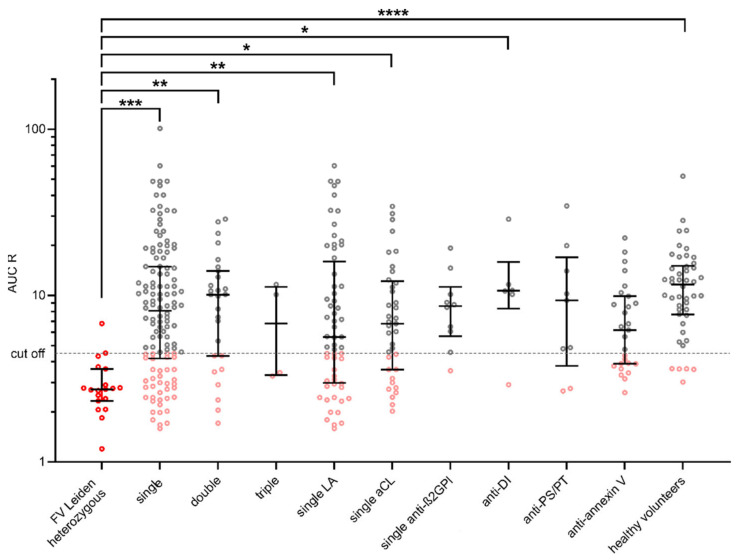
Visualization of AUC R in each group by scatter dot plots including median and quartiles. Cut-off for AUC R ≤ 4.5 (90th percentile of AUC R in patients with FV Leiden heterozygous). TGA: thrombin generation assay. AUC R: ratio of area under the curve AUC − APC/AUC + APC. On the basis of this cut-off value, patients were assessed as high- or low-thrombogenic. Statistically significant differences between groups using ANOVA are shown above the graph (*, **, *** and **** for *p* values <0.05, <0.01, <0.001, and <0.0001, respectively). The group of FV Leiden heterozygotes as a standard of thrombogenicity is shown by the red shaded points. Values below the cut-off for each antibody type are shown in pink.

**Table 1 ijms-23-08973-t001:** Overview of population in the examined patient cohort. APS: antiphospholipid syndrome. SN-APS: seronegative antiphospholipid syndrome.

	APS	SN-APS	Healthy Volunteers	FV Leiden Heterozygous
N	139	36	47	21
Sex (women/men)	70/69	14/22	34/13	8/13

**Table 2 ijms-23-08973-t002:** Overview of biomarkers in the examined patient cohort. APS: antiphospholipid syndrome; SN-APS: seronegative antiphospholipid syndrome; LA: lupus anticoagulant; aCL: anti-anticardiolipin; anti-β2GPI: anti-beta-2-glycoprotein I; anti-DI: anti-beta-2-glycoprotein I Domain I; anti-PS/PT: anti-phosphatidylserine/prothrombin.

	APS	SN-APS
Single positivity	108/139	
Double positivity	27/139	
Triple positivity	4/139	
Single positivity LA	61/139	
Single positivity aCL	37/139	
Single positivity anti-β2GPI	10/139	
anti-DI	6/139	
Anti-PS/PT		9/36
Anti-annexin V		27/36

**Table 3 ijms-23-08973-t003:** Summary of the representation of patients with a cut-off ≤ 4.5 for AUC R in each group. LA: lupus anticoagulant; aCL: anti-anticardiolipin; anti-β2GPI: anti-β2-glycoprotein I; anti-PS/PT: anti-phosphatidylserine/prothrombine; anti-DI: anti-beta-2-glycoprotein I Domain I.

	Cut-Off ≤ 4.5	
	% (N)	*p* Value
Single positivity	35.2 (38)	<0.001
Double positivity	29.6 (8)	<0.01
Triple positivity	50.0 (2)	
Single positivity LA	41.0 (25)	<0.01
Single positivity aCL	32.4 (12)	<0.05
Single positivity anti-β2GPI	10.0 (1)	
Anti-annexin V	40.7 (11)	
Anti-PS/PT	22.2 (2)	
anti-DI	16.7 (1)	<0.05
Healthy volunteers	10.6 (5)	<0.0001

## Data Availability

All data generated in this study are included in this published article.

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
