# Peer review of "Determination of Thrombogenicity Levels of Various Antiphospholipid Antibodies by a Modified Thrombin Generation Assay in Patients with Suspected Antiphospholipid Syndrome"

_ijms, 2022, doi:10.3390/ijms23168973_

Round 1

Reviewer 1 Report

This manuscript is ill-prepared and contains various incorrect, inconsistent or inaccurate statements and presentations.

Abstract:

The abstract is quite wordy yet does not provide the rationale of the study design. It should be made more concise and focused. Most critically, its conclusions are not supported by the results presented.

Lines 42-50 “Our study demonstrates the well-known fact that multiple positivity of different aPLs is a more severe risk for thrombosis than single positivity. Of the single antibody positivity, LA antibodies are the most serious. However, even anti-annexin V positivity achieves similar AUC R values as LA antibodies in patients, which contrasts with the low values achieved with anti-B2GPI1 positivity.”

Without statistical analysis for difference, these statements are unwarranted.

Introduction:

Instead of proving the rationale for the study, the introduction section is a lengthy review of the definition of APS and the current status of its laboratory diagnosis that is not directly relevant to this study.

This study uses the APC resistance of thrombin generation as the surrogate marker of thrombogenicity. The literature that is relevant to this premise should be elaborated in the Introduction section.

Study designs:

The study sample numbers do not match.

Lines 182-186 “We examinated 243 samples, of these were 176 patients with clinical manifestations of APS according to Sydney criteria classification, 47 healthy volunteers as negative control group, and 21 patients with positive activated protein C resistance in heterozygous carriers of the Leiden FV mutation (FV Leiden heterozygous) as positive control group were included in the retrospective study to determine the cut-off.”

Firstly, “examinated” is not an English word.

Secondly, 176+47+21 equals 244, not 243. In Table 1, there are 139+36=175 cases, not 176 cases, of APS. The inconsistency should be corrected.

Thirdly, the statement “….176 patients with clinical manifestations of APS according to Sydney criteria classification.…” is unclear and can be misleading. It should be revised to be more precise.

Lines 193-194 “In patients taking direct oral anticoagulants (DOACs), drug elimination from plasma was performed with a DOAC stop tablet [47].”

The possibility of clinical conditions and DOAC treatment affecting the endogenous thrombin generation potential of the coagulation system should be elaborated.

Materials and Methods:

The precisions of TGA parameters and their respective APC ratios should be delineated. The impact of clinical conditions under which the blood samples were collected on the TGA should also be examined.

Lines 212-213  For interpretation LA results was used cut-off value of normalized ratio (NR = patient/polled normal plasma) NR >1,2. [51].

The sentence is unintelligible.

Lines 219-220 “A compatible human Ig, labelled with isoluminol, binds to this resulting complex.”

This is a misrepresentation of the test.

Lines 223-226 “The measured RLUs are directly proportional to the concentration of each aPL in the sample. Through the 4PLC logistic curve, the measured RLUs are converted to chemiluminescence units (CU), the cut-off recommended by the manufacturer is >20 U/ml [54].”

The first sentence is not supported by the cited reference. “4PLC logistic curve” should be defined. The sentence is also grammatically unacceptable.

Line 234 “A conjugate of human Ig and peroxidase is bound to this complex.”

This is factually incorrect. 

Lines 244-247 “In contrast to basic coagulation assays such as Prothrombin time and aPTT, which measure thrombin generation only during the initiation phase of coagulation, TGA measures thrombin generation during the entire clotting process and thus provides a broader picture of the overall coagulation potential.”

This characterization of TGA does not belong to the “Materials and Methods” section. Instead, the reagent that was used to trigger thrombin generation should be mentioned.

Lines 256-257 “The risk of bleeding or thrombosis can be predicted based on TGA testing [59, 60].”

Neither of the two citations presents or mentions studies with the data that support the “risk of bleeding or thrombosis” claim.

Lines 258-260 “In our study, for each PPP sample, one TGA measurement was performed with reagent with added activated protein C (+APC) and a second measurement without added activated protein C (-APC).”

The source and concentration of APC should be indicated.

Line 265 “For normalize original data log-transformation was applied”

The sentence is grammatically incorrect.

Results (Figures and Table):

The boxplots should be clearly defined.

Why is the 75% percentile different from the third quartile on the plot?

Part of the Figure 1 legend is confusing and does make sense: “………This section may be divided by subheadings. It should provide a concise and precise description of the experimental results, their interpretation, as well as the experimental conclusions that can be drawn.”

Consistency: Are “tLag R” and “AUC R” in the legends and “R(t_lag)” and R(AUC) in the y-axis labels the same?

The data presented in the figures and table should be statistically analyzed for difference.

In Table 1, the denominators for the fractions should be included. Do they add up to 175?

Discussion:

The discussion should be completely revised based on the results of statistical analysis. It should also be more concise and focused.

References:

This section needs a thorough proofread for inconsistency, inaccuracy and incorrectness.

Author Response

Abstract:

The abstract is quite wordy yet does not provide the rationale of the study design. It should be made more concise and focused. Most critically, its conclusions are not supported by the results presented.

The abstract was completely revised.

Lines 42-50 “Our study demonstrates the well-known fact that multiple positivity of different aPLs is a more severe risk for thrombosis than single positivity. Of the single antibody positivity, LA antibodies are the most serious. However, even anti-annexin V positivity achieves similar AUC R values as LA antibodies in patients, which contrasts with the low values achieved with anti-B2GPI1 positivity.”

Without statistical analysis for difference, these statements are unwarranted.

We did a statistical evaluation and adjusted our conclusions.

Introduction:

Instead of proving the rationale for the study, the introduction section is a lengthy review of the definition of APS and the current status of its laboratory diagnosis that is not directly relevant to this study.

This study uses the APC resistance of thrombin generation as the surrogate marker of thrombogenicity. The literature that is relevant to this premise should be elaborated in the Introduction section.

The introduction was completely revised.

Study designs:

The study sample numbers do not match.

Lines 182-186 “We examinated 243 samples, of these were 176 patients with clinical manifestations of APS according to Sydney criteria classification, 47 healthy volunteers as negative control group, and 21 patients with positive activated protein C resistance in heterozygous carriers of the Leiden FV mutation (FV Leiden heterozygous) as positive control group were included in the retrospective study to determine the cut-off.”

Firstly, “examinated” is not an English word.

We corrected it - detected

Secondly, 176+47+21 equals 244, not 243. In Table 1, there are 139+36=175 cases, not 176 cases, of APS. The inconsistency should be corrected.

The number 176 was misspelled, the correct number is 175 J.

Thirdly, the statement “….176 patients with clinical manifestations of APS according to Sydney criteria classification.…” is unclear and can be misleading. It should be revised to be more precise.

Patients (175) who had a proven history of thrombosis or fetal loss were recruited. We wrote it: We examined 243 samples, of these were 175 patients with clinical manifestations of APS according to Sydney criteria classification, that is, each patient had thrombosis or fetal loss and was positive for at least one aPL: LA, aCL, anti-β2GP1.

Lines 193-194 “In patients taking direct oral anticoagulants (DOACs), drug elimination from plasma was performed with a DOAC stop tablet [47].”

The possibility of clinical conditions and DOAC treatment affecting the endogenous thrombin generation potential of the coagulation system should be elaborated.

We have inserted a source to support this claim: Slavik et al conducted a study to verify the effect of DOAC stop tablets and found that after elimination of the DOAC drug from the plasma sample, neither screening nor special coagulation tests were affected. 

Materials and Methods:

The precisions of TGA parameters and their respective APC ratios should be delineated. The impact of clinical conditions under which the blood samples were collected on the TGA should also be examined.

In Harper et al. the mechanism of protein C in relation to the action of antiphostolipid antibodies is described, where Protein C (PC) is activated by thrombin bound to thrombomodulin and acts as a key regulator of thrombosis by inactivating FVa and FVIIa and activating PAR1, which leads to the inhibition of apoptosis. Modulation of aPC in some APS patients who have increased resistance to aPC leads to greater thrombin generation. The presence of in vitro aPC resistance is associated with the presence of LAC and a clinical history of thrombosis.

We used this in our work to modify the TGT assay by adding APC and assessed the effect of the modification on patients with antiphospholipid antibodies, which appears to be very appropriate from the results, mainly for two reasons. We can assess the effect of antibodies on a known target causing thrombotic manifestations, by which I APC and the other , avoiding inter-individual differences caused by sampling timing and other influences, as we evaluate the ratio of tests.

Harper BE, Wills R, Pierangeli SS. Pathophysiological mechanisms in antiphospholipid syndrome. Int J Clin Rheumtol. 2011 Apr 1;6(2):157-171. doi: 10.2217/ijr.11.9. PMID: 23487578; PMCID: PMC3593246.

Curvers J, Christella M, Thomassen LG, de Ronde H, Bertina RM, Rosendaal FR, Tans G, Rosing J. Effects of (pre-)analytical variables on activated protein C resistance determined via a thrombin generation-based assay. Thromb Haemost. 2002 Mar;87(3):483-92. PMID: 11924538.

Lines 212-213  “For interpretation LA results was used cut-off value of normalized ratio (NR = patient/polled normal plasma) NR >1,2. [51]. “

The sentence is unintelligible.

We wrote new explanation: The LA results were interpreted as positive/negative according to the ISTH SSC recommendations based on the calculation of the normalized ratio (NR). The calculation of the NR is as follows: NR = R screening (time patient/ time polled normal plasma) / R confirmation (time patient/ time polled normal plasma. The cut-off value for NR is 1.2. Samples that exceeded the cut-off >1.2 were determined to be LA positive.

Lines 219-220 “A compatible human Ig, labelled with isoluminol, binds to this resulting complex.”

This is a misrepresentation of the test.

Lines 223-226 “The measured RLUs are directly proportional to the concentration of each aPL in the sample. Through the 4PLC logistic curve, the measured RLUs are converted to chemiluminescence units (CU), the cut-off recommended by the manufacturer is >20 U/ml [54].”

The first sentence is not supported by the cited reference. “4PLC logistic curve” should be defined. The sentence is also grammatically unacceptable.

We deleted this interpretations and replaced by new sentence: Measurements were performed according to the reagent kit manufacturer's recommendation.

Line 234 “A conjugate of human Ig and peroxidase is bound to this complex.”

This is factually incorrect. 

We deleted this interpretation and replaced by new sentence: Measurements were performed according to the reagent kit manufacturer's recommendation.

Lines 244-247 “In contrast to basic coagulation assays such as Prothrombin time and aPTT, which measure thrombin generation only during the initiation phase of coagulation, TGA measures thrombin generation during the entire clotting process and thus provides a broader picture of the overall coagulation potential.”

This characterization of TGA does not belong to the “Materials and Methods” section.

We replaced this to the Introduction.

Instead, the reagent that was used to trigger thrombin generation should be mentioned.

Lines 256-257 “The risk of bleeding or thrombosis can be predicted based on TGA testing [59, 60].”

Neither of the two citations presents or mentions studies with the data that support the “risk of bleeding or thrombosis” claim.

Yes, we agree, there is no cut-off for predicting thrombosis or bleeding in the two referenced studies, however, TGA appears to be a promising tool in these cases.

We have added a new sentence: The risk of bleeding or thrombosis can be predicted based on TGA testing. Based on an analysis of the literature, Tripodi et al concluded that TGA is a promising laboratory tool for investigating bleeding coagulopathies and predicting the risk of recurrent venous thromboembolism.

Lines 258-260 “In our study, for each PPP sample, one TGA measurement was performed with reagent with added activated protein C (+APC) and a second measurement without added activated protein C (-APC).”

The source and concentration of APC should be indicated.

We have inserted concentration both reagents: Thrombin generation in the sample was initiated with 7.16 pM of recombinant tissue factor (rTF) resuspended in 0.32 μM of phospholipid micelles (containing 2.56 μM of phosphatidylcholineand 0.64 μM of phosphatidylserine). For APC, activated protein C (16 nmol/l) was added.

Line 265 “For normalize original data log-transformation was applied”

The sentence is grammatically incorrect.

We corrected it completely: Statistical analysis was performed using the software GraphPad Prism 9.0 (GraphPad Software, San Diego, CA, USA). To normalize original (tLag R) data log-transformation was applied. Consequently, Shapiro-Wilk test confirmed the gaussian distribution of all groups studied. Outliers were identified by Grubb’s test (alpha = 0.05). Data were visualised by scatter dot plots including median and quartiles. ANOVA with Dunnett's multiple comparisons post-hoc test was applied to calculate statistical significance of differences between FV Leiden heterezygous and other groups. Software GraphPad 9.0 was used for all statistical evaluations.

Results (Figures and Table):

The boxplots should be clearly defined.

Why is the 75% percentile different from the third quartile on the plot?

Part of the Figure 1 legend is confusing and does make sense: “………This section may be divided by subheadings. It should provide a concise and precise description of the experimental results, their interpretation, as well as the experimental conclusions that can be drawn.”

Consistency: Are “tLag R” and “AUC R” in the legends and “R(t_lag)” and R(AUC) in the y-axis labels the same?

We have created a new, clearer chart. We have marked the points below the cut-off in pink for clarity, removed boxes and added statistical evaluation between studied groups:

We corrected description of the figure 2: Visualisation of AUC R in each group by scatter dot plots including median and quartiles. Cut-off for AUC R ≤ 4.5 (90th percentile of AUC R in patients with FV Leiden heterozygous). TGA: Thrombin generation assay. AUC R: ratio of area under the curve AUC-APC/ AUC+APC. On basis this cut-off value was assessed patient as high and low thrombogenic. Statistically significant differences between groups using ANOVA are shown above the graph (*, **, *** and **** for p values <0.05, <0.01, <0.001 and <0.0001, respectively).

We have unified the description of the parameters in the graph and in the text.

The data presented in the figures and table should be statistically analyzed for difference.

In Table 1, the denominators for the fractions should be included. Do they add up to 175?

We have divided the data in the table into two new tables for better clarity. We applied ANOVA to evaluate differences between studied groups.

Discussion:

The discussion should be completely revised based on the results of statistical analysis. It should also be more concise and focused.

We have edited the discussion.

References:

This section needs a thorough proofread for inconsistency, inaccuracy and incorrectness.

We used EndNote - reference management tool.

Reviewer 2 Report

Major comments:

11.)    The manuscript provided is not structured accordingly and raises a lot of questions both

a.       The introduction is too general and does not discuss antibody specificity

b.      The introduction lacks insights on the isotype and subclasses of all aPL antibodies

c.       What are the mechanisms of thrombogenicity?

d.      What are the methods to evaluate risk?

e.       Is the sample size large enough to outline statistical relevance?

22.)    What are the actual research questions and how will the authors use methods – test systems – to address this complicated issue?

33.)     Is the modified test system robust enough to allow a conclusion based on one method? As the authors use only one test system, how can they be sure that they have the right method in hand? I would prefer to see an additional assay system in addition not to relay on one.

44.)    Study design and population/patients & healthy individuals should belong to the Methods

55.)    In the methods, the authors describe the principles of commercially available systems. Please refer as according to the manufacturer’s recommendation.

66.)    The very basic Technothrombin TGA assay by Technoclone is modified by the addition of APC and a ratio is calculated. The rational for this calculation is not given, instead a preliminary study is cited. Please explain in detail what the benefit is to modify the commercially available assay. I also recommend the use of another commercial assay with the same specificity to increase evidence on the robustness of the assay used.

77.)    Looking at Figure 1, I cannot see a difference between the groups. There is a great spread, with scattering over the range of detection in almost all groups investigated.

88.)    Figure 2 is logarithmic this time and shows the same scattering. Here I also cannot detect a trend.

99.)    From the results I cannot follow the conclusions presented in the discussion.

Minor comments:

11.)      There are several authors to improve the paper, adopt to a more mechanistic than descriptive fashion of data presentation and develop the scientific content of the manuscript adequately to meet the high requirements of IJMS.

22.)      In principle, the manuscript better fits to a more clinical journal since no molecular mechanisms of antibody specificities are addressed. Binding strength, Ka, Kd, and avidity are what comes up in my mind.

Author Response

Major comments:

1.) The manuscript provided is not structured accordingly and raises a lot of questions both

a. The introduction is too general and does not discuss antibody specificity

The different antibodies investigated in APS (LA, aCL, anti-β2GPI, anti-prothrombin (anti-PT), anti-phosphatidylserine/prothrombin (anti-PS/PT), and anti-phosphatidylethanolamine) can be evaluated for specificity of phospholipid surface inhibition. However, laboratory evaluation of antibody manifestations is difficult and thus most studies are unfortunately based on testing the clinical manifestation APS of a specific antibody.   The combination of LA + anti-β2GPI + anti-PS/PT provides the best diagnostic accuracy for APS, according to Scascia [16].

Sciascia S, Murru V, Sanna G, Roccatello D, Khamashta MA, Bertolaccini ML. Clinical accuracy for diagnosis of antiphospholipid syndrome in systemic lupus erythematosus: evaluation of 23 possible combinations of antiphospholipid antibody specificities. J Thromb Haemost. 2012 Dec;10(12):2512-8.

b. The introduction lacks insights on the isotype and subclasses of all aPL antibodies

From the point of view of the antibody isotype, the most important for the clinical manifestation of PAS are LA with anticoagulant activity, in contrast to the isolated aCL and anti-β2GPI in the IgG and IgM classes. The importance of the IgA class is also rarely described.

c. What are the mechanisms of thrombogenicity?

Regarding the pathophysiology, a number of possible activation mechanisms of antiphospholipid antibodies against cellular targets (monocytes, platelets, endothelial cells) have been described, however, APCs appear to be the main target in the humoral part. Protein C (PC) is activated by thrombin bound to thrombomodulin and serves as a key regulator of thrombosis by inactivating FVa and FVIIa and activating PAR1, which leads to the inhibition of apoptosis. Modulation of aPC increased resistance to aPC and  increased thrombin generation. The presence of in vitro aPC resistance is associated with the presence of LAC and a clinical history of thrombosis.

d. What are the methods to evaluate risk?

The pathogenic mechanisms by which aPL induce disease expression in APS are manifold and the corollary of this is the exceptional variability in the clinical manifestations observed in these patients. Unfortunately, the molecular mechanisms that lead to aPL development and aPL-induced disease expression remain incompletely understood. Methods that can be used to detect these changes are molecular-based, flowcytometric, immunological and functional. However, they do not fully reflect the in-vivo action of antibodies on target molecules and need to be complemented by clinical outcom.

Groot PG, Derksen RH. Pathophysiology of the antiphospholipid syndrome. J Thromb Haemost. 2005

e. Is the sample size large enough to outline statistical relevance?

We calculated Power of the study and added this sentence to method part: „The statistical significance of the study (power analysis) was calculated as Cohen's D value (statistically important minimum effect size) in the range 0.68 – 1.61 for the comparison of FV Leiden heterozygous samples (N=21) with other studied groups (N=4 – 108) for target parameters of power = 0.8 and alpha = 0.05.“

2.) What are the actual research questions and how will the authors use methods – test systems – to address this complicated issue?

We replaced this to The Introduction.

3.) Is the modified test system robust enough to allow a conclusion based on one method? As the authors use only one test system, how can they be sure that they have the right method in hand? I would prefer to see an additional assay system in addition not to relay on one.

In Harper et al. the mechanism of protein C in relation to the action of antiphostolipid antibodies is described, where Protein C (PC) is activated by thrombin bound to thrombomodulin and acts as a key regulator of thrombosis by inactivating FVa and FVIIa and activating PAR1, which leads to the inhibition of apoptosis. Modulation of aPC in some APS patients who have increased resistance to aPC leads to greater thrombin generation. The presence of in vitro aPC resistance is associated with the presence of LAC and a clinical history of thrombosis.

We used this in our work to modify the TGT assay by adding APC and assessed the effect of the modification on patients with antiphospholipid antibodies, which appears to be very appropriate from the results, mainly for two reasons. We can assess the effect of antibodies on a known target causing thrombotic manifestations, by which I APC and the other , avoiding inter-individual differences caused by sampling timing and other influences, as we evaluate the ratio of tests.

Harper BE, Wills R, Pierangeli SS. Pathophysiological mechanisms in antiphospholipid syndrome. Int J Clin Rheumtol. 2011 Apr 1;6(2):157-171. doi: 10.2217/ijr.11.9. PMID: 23487578; PMCID: PMC3593246.

Curvers J, Christella M, Thomassen LG, de Ronde H, Bertina RM, Rosendaal FR, Tans G, Rosing J. Effects of (pre-)analytical variables on activated protein C resistance determined via a thrombin generation-based assay. Thromb Haemost. 2002 Mar;87(3):483-92. PMID: 11924538.

4.) Study design and population/patients & healthy individuals should belong to the Methods

We replaced this to the Methods.

5.) In the methods, the authors describe the principles of commercially available systems. Please refer as according to the manufacturer’s recommendation.

We deleted this interpretations and replaced by new sentence: Measurements were performed according to the reagent kit manufacturer's recommendation.

6.) The very basic Technothrombin TGA assay by Technoclone is modified by the addition of APC and a ratio is calculated. The rational for this calculation is not given, instead a preliminary study is cited. Please explain in detail what the benefit is to modify the commercially available assay. I also recommend the use of another commercial assay with the same specificity to increase evidence on the robustness of the assay used.

In Harper et al. the mechanism of protein C in relation to the action of antiphostolipid antibodies is described, where Protein C (PC) is activated by thrombin bound to thrombomodulin and acts as a key regulator of thrombosis by inactivating FVa and FVIIa and activating PAR1, which leads to the inhibition of apoptosis. Modulation of aPC in some APS patients who have increased resistance to aPC leads to greater thrombin generation. The presence of in vitro aPC resistance is associated with the presence of LAC and a clinical history of thrombosis.

We used this in our work to modify the TGT assay by adding APC and assessed the effect of the modification on patients with antiphospholipid antibodies, which appears to be very appropriate from the results, mainly for two reasons. We can assess the effect of antibodies on a known target causing thrombotic manifestations, by which I APC and the other , avoiding inter-individual differences caused by sampling timing and other influences, as we evaluate the ratio of tests.

Harper BE, Wills R, Pierangeli SS. Pathophysiological mechanisms in antiphospholipid syndrome. Int J Clin Rheumtol. 2011 Apr 1;6(2):157-171. doi: 10.2217/ijr.11.9. PMID: 23487578; PMCID: PMC3593246.

Curvers J, Christella M, Thomassen LG, de Ronde H, Bertina RM, Rosendaal FR, Tans G, Rosing J. Effects of (pre-)analytical variables on activated protein C resistance determined via a thrombin generation-based assay. Thromb Haemost. 2002 Mar;87(3):483-92. PMID: 11924538.

7.) Looking at Figure 1, I cannot see a difference between the groups. There is a great spread, with scattering over the range of detection in almost all groups investigated.

8.) Figure 2 is logarithmic this time and shows the same scattering. Here I also cannot detect a trend.

We have created a new, clearer scatter plots. We corrected desription of y-axis. In the figure 2 just logarithm scale was applied on original data.

We have marked the points below the cut-off in red for clarity, removed box-plots and added statistical significance groups:

Visualisation of AUC R in each group by scatter dot plots including median and quartiles. Cut-off for AUC R ≤ 4.5 (90th percentile of AUC R in patients with FV Leiden heterozygous). TGA: Thrombin generation assay. AUC R: ratio of area under the curve AUC-APC/ AUC+APC. On basis this cut-off value was assessed patient as high and low thrombogenic. Statistically significant differences between groups using ANOVA are shown above the graph (*, **, *** and **** for p values <0.05, <0.01, <0.001 and <0.0001, respectively).

We have unified the description of the parameters in the graph and in the text.

9.) From the results I cannot follow the conclusions presented in the discussion.

We updated Figure 2 for better vizualization of the data and added statistical evaluation based on ANOVA.

Minor comments:

1.) There are several authors to improve the paper, adopt to a more mechanistic than descriptive fashion of data presentation and develop the scientific content of the manuscript adequately to meet the high requirements of IJMS.

2.) In principle, the manuscript better fits to a more clinical journal since no molecular mechanisms of antibody specificities are addressed. Binding strength, Ka, Kd, and avidity are what comes up in my mind.

Round 2

Reviewer 2 Report

The submitted manuscript entitled ‘Determination of thrombogenicity levels various antiphospholipid antibodies by a modified thrombin generation assay in patients with suspected antiphospholipid syndrome’, by Pavla Bradacova et al, has been revised but still lacks important issues.

Major comments:

1.)    It has not been addressed adequately, whether the modified test system is robust enough to allow the conclusion presented in the manuscript. As the authors use only one test system, how can they be sure that they have the right method in hand? I would prefer to see an alternative assay system.

2.)    The very basic Technothrombin TGA assay by Technoclone is modified by the addition of APC and a ratio is calculated. The rational for this calculation is still not clear to me. Please explain in detail what the benefit is to modify the commercially available assay. I also recommend the use of another commercial assay with the same specificity to increase evidence on the robustness of the assay used.

Minor comments:

In principle, the manuscript better fits to a more clinical journal since no molecular mechanisms of antibody specificities are addressed that are state of the art today. Binding strength, Ka, Kd, and avidity are parameters that would add a lot to the outcome.

Author Response

1. It has not been addressed adequately, whether the modified test system is robust enough to allow the conclusion presented in the manuscript. As the authors use only one test system, how can they be sure that they have the right method in hand? I would prefer to see an alternative assay system.

Jonathan Douxfils in work “Validation and standardization of the ETP-based activated protein C resistance test for the clinical investigation of steroid contraceptives in women: an unmet clinical and regulatory need” described the validation parameters of TGA method modified by APC. Our results has similar intra a inter run repeatability with maximum 7,2 % for inter run repeatability for low control.

We added information about it to Introduction part (Line 113-117): “This property of TGA analysis can be used to detect various causes of hypercoagulable states, when Douxfils et al [18] used a modification of TGT with the addition of APC to measure the effect of hormonal contraceptives, it is also possible to use this modified test to determine the influence of phospholipid antibodies to APC.”

Wayne Chandler in work “Optimization of Plasma Fluorogenic Thrombin-Generation” compared both available TGT tests (Thrombinoscope now Stago and Technoclone) and demonstrated their comparability. In the work, the tests are compared at different concentrations of TF and phospholipids, and the variability of sampling and its stabilization by corn trypsin inhibitor, which appeared to be a problem but does not concern us, is also compared, as we are simultaneously comparing only the ratio and not the absolute value.

We added information about comparability of the methods to Discussion part (Line 380-390): “The modified TGA method with the addition of APC was used to detect antiphospholipid antibodies as in the work of Douxfils et al. (18), where it was used to determine the thrombogenicity of hormonal contraceptives. Our results show comparable interassay variability. On the basis of previously published results comparing the two available TGA methods (Wayne Chandler et al.59), our results can be advantageously adapted to a second routinely used testing platform.”

2. The very basic Technothrombin TGA assay by Technoclone is modified by the addition of APC and a ratio is calculated. The rational for this calculation is still not clear to me. Please explain in detail what the benefit is to modify the commercially available assay. I also recommend the use of another commercial assay with the same specificity to increase evidence on the robustness of the assay used.

Jonathan Douxfils in work “Validation and standardization of the ETP-based described using modified method for detecting thrombogenicity oral contraceptives that is on basis unclear mechanism of action, but directed against the APC system (probably based on an increase in the levels of procoagulant factors). We used this robustly validated test to determine the effect of phospholipid antibodies on this system.

We added information about importance of ratio in the modeified TGA method to Discussion part (Line 385-391): “Expression of the results of the modified TGA method as a ratio R=TGA-APC/ TGA+APC provides the possibility to evaluate the change in thrombin generation depending on the presence of APC and avoids the influence of interindividual variability caused by the presence of different types of antibodies. At the same time, by measuring the ratios, we avoid possible assay variability caused by the time factor of collection and other laboratory variables (59).”

1.) There are several authors who wish to improve the article, adopt a more mechanistic than descriptive way of presenting the data, and develop the scientific content of the manuscript adequately to meet the high requirements of the IJMS.

We believe that we sufficiently describe the methodology for determining the functional manifestation of the presence of individual types of antiphospholipid antibodies, which corresponds to their clinical manifestation, which is essential for evaluating the effect of antibodies in the pathophysiological process of blood clotting inhibition. It was not the goal of our work to mechanoscopically describe antibodies and their structure.

2.) Basically, the manuscript is better suited to a more clinical journal because it does not deal with any molecular mechanisms of antibody specificities. Bond strength, Ka, Kd and avidity are what comes to mind.

We fully accept your comment. However, the determination of basic antibody specificities does not clarify the consequence of the presence of antibodies, their binding strength, Ka, Kd described in the literature. For this reason, we present our data based on the manifestation of antibodies against one of the strongest inhibitory systems of coagulation, which is the anti-C system with a known manifestation in thrombotic complications. In this way, we can help in the identification of patients with manifest antibodies that cause the formation of thrombotic complications.
